# CAN REINFORCEMENT LEARNING EFFICIENTLY FIND STACKELBERG-NASH EQUILIBRIA IN GENERAL-SUM MARKOV GAMES?

## ABSTRACT

We study multi-player general-sum Markov games with one of the players designated as the leader and the rest regarded as the followers. In particular, we focus on the class of games where the state transitions are only determined by the leader's action while the actions of all the players determine their immediate rewards. For such a game, our goal is to find the Stackelberg-Nash equilibrium (SNE), which is a policy pair $(\pi^*, \nu^*)$ such that (i) $\pi^*$ is the optimal policy for the leader when the followers always play their best response, and (ii) $\nu^*$ is the best response policy of the followers, which is a Nash equilibrium of the followers' game induced by $\pi^*$. We develop sample efficient reinforcement learning (RL) algorithms for solving SNE for both the online and offline settings. Respectively, our algorithms are optimistic and pessimistic variants of least-squares value iteration and are readily able to incorporate function approximation for handling large state spaces. Furthermore, for the case with linear function approximation, we prove that our algorithms achieve sublinear regret and suboptimality under online and offline setups respectively. To our best knowledge, we establish the first provably efficient RL algorithms for solving SNE in general-sum Markov games with leader-controlled state transitions.

## 1 INTRODUCTION

Reinforcement learning (RL) has achieved striking empirical successes in solving complicated real-world sequential decision-making problems (Mnih et al., 2015; Duan et al., 2016; Silver et al., 2016; 2017; 2018; Agostinelli et al., 2019; Akkaya et al., 2019). Motivated by these successes, multi-agent extensions of RL algorithms recently have gained great popularity in decision-making problems involving multiple interacting agents (Busoniu et al., 2008; Hernandez-Leal et al., 2018; 2019; Oroo-jlooyJadid & Hajinezhad, 2019; Zhang et al., 2019). Multi-agent RL is often modeled as a Markov game (Littman, 1994) where, at each time step, each player (agent) takes an action simultaneously at each state of the environment, observe her own immediate reward, and the environment evolves into a next state. Here both the reward of each player and the state transition depends on the actions of all players. From the perspective of each player, her goal is to find a policy that maximizes her expected total reward in the presence of other agents.

In Markov games, depending on the structure of the reward functions, the relationship among the players can be either collaborative, where each player has the same reward function, or competitive, where the sum of the reward function is equal to zero, or mixed, which corresponds to a general-sum game. While most of existing theoretical results focus on the collaborative or two-player competitive settings, the mixed setting is oftentimes more pertinent to real-world multi-agent applications.

Moreover, in addition to having diverse reward functions, the players might also have asymmetric roles in the Markov game — the players might be divided into leaders and followers, where the leaders' joint policy determines a general-sum game for the followers. Games with such a leader-follower structure is popular in applications such as mechanism design (Conitzer & Sandholm, 2002; Roughgarden, 2004; Garg & Narahari, 2005; Kang & Wu, 2014), security games (Tambe, 2011; Korzhyk et al., 2011; Balcan et al., 2015), incentive design (Zheng et al., 1984; Ratliff et al., 2014; Chen et al., 2016; Ratliff & Fiez, 2020), and model-based RL (Rajeswaran et al., 2020). Consider a simplified economic system that consists of a government and a group of companies, where the

companies purchase or sell goods, and the government collects taxes from transactions. Such a problem can be viewed as a multi-player general-sum game, where the government serves as the leader and the companies are followers (Zheng et al., 2020). In particular, when the government sets a tax rate, the companies form a general-sum game themselves, whose reward functions depend on the tax rate. Each company aims to maximize their own revenue, and thus ideally they achieve a Nash equilibrium (NE) of the induced game. Whereas the goal of the government might be achieving the social welfare, which might be measured via certain fairness metrics computed by the revenues of the companies.

In multi-player Markov games with such a leader-follower structure, the desired solution concept is the Stackelberg-Nash equilibrium (SNE) (Başar & Olsder, 1998). In the setting where there is a single leader, SNE corresponds to a pair of leader's policy $\pi^*$ and followers' joint policy $\nu^*$ that satisfies the following two properties: (i) when the leader adopts $\pi^*$, $\nu^*$ is the best-response policy of the followers, i.e., $\nu^*$ is a Nash equilibrium of the followers' subgame induced by $\pi^*$; and (ii) $\pi^*$ is the optimal policy of the leader assuming the followers always adopt the best response.

We are interested in finding an SNE in a multi-player Markov game when the reward functions and Markov transition kernel are unknown. In particular, we focus on the setting with a single leader and the state transitions only depend on the leader's actions. That is, the followers' actions only affect the rewards received by the leader and followers. For such a game, we are interested in the following question:

Can we develop reinforcement learning methods that provably find Stackelberg-Nash equilibria in leader-controlled general-sum games with sample efficiency?

To this end, we consider both online and offline RL settings, where in the former, we learn the SNE in a trial-an-error fashion by interacting with the environment and generating data, and in the latter, we learn the SNE from a given dataset that is collected a priori. For the online setting, as the transition model is unknown, to achieve sample efficiency, the equilibrium-finding algorithm also needs to take the exploration-exploitation tradeoff into consideration. Although the similar challenge has been studied in zero-sum Markov game, it seems unclear how to incorporate popular exploration mechanisms such as optimism in the face of uncertainty (Sutton & Barto, 2018) into SNE finding. Meanwhile, under the offline setting, as the RL agent has no control of data collection, it is ideal to design an RL algorithm with theoretical guarantees for an arbitrary dataset that might not be sufficiently explorative.

**Our contributions**    Our contributions are three-fold. First, for the episodic leader-controlled general-sum game, under the online and offline settings respectively, we propose optimistic and pessimistic variants of the least-squares value iteration (LSVI) algorithm. In particular, in a version of LSVI, we estimate the optimal action-value function of the leader via least-squares regression and construct an estimate of the SNE by solving the SNE of the multi-matrix game for each state, whose payoff matrices are given by the leader's estimated action-value function and the followers' reward functions. Moreover, we add a UCB exploration bonus to the least-squares solution to achieve optimism in the online setting. Whereas in the offline setting, pessimism is achieved by subtracting a penalty function constructed using the offline data, which is equal to the negative bonus function. Moreover, these algorithms are readily able to incorporate function approximators and we showcase the version with linear function approximation. Second, under the online setting, we prove that our optimistic LSVI algorithm achieves a sublinear $\widetilde{\mathcal{O}}(H^2\sqrt{d^3 K})$ regret, where $K$ is the number of episodes, $H$ is the horizon, $d$ is the dimension of the feature mapping, and $\widetilde{\mathcal{O}}(\cdot)$ omits logarithmic terms. Finally, under the offline setting, we establish an upper bound on the suboptimality of the proposed algorithm for an arbitrary dataset with $K$ trajectories. Our upper bound yields a sublinear $\widetilde{\mathcal{O}}(H^2\sqrt{d^3/K})$ rate as long as the dataset has sufficient coverage over the trajectory induced by the desired SNE.

**Related work**    In the sequel, we discuss the related works on learning Stackelberg games. We defer more related works on RL for solving NE in Markov games and single-agent RL to §A.

**Learning Stackelberg games**    As for solving Stackelberg-Nash equilibrium, most of the existing results focus on the normal form game, which is equivalent to our Markov game with $H = 1$. Letchford et al. (2009); Blum et al. (2014); Peng et al. (2019) study learning Stackelberg equilibrium with a best response oracle. In addition, Fiez et al. (2019) study the local convergence of first-order methods for finding Stackelberg equilibria in general-sum games with differentiable reward functions, and Ghadimi & Wang (2018); Chen et al. (2021a); Hong et al. (2020) analyze the global convergence of first-order methods for achieving global optimality of bilevel optimization. A more

related work is Bai et al. (2021), which studies the matrix Stackelberg game with bandit feedback. This work also studies an RL extension where the leader has a finite action set and the follower is faced with an MDP specified by the leader's action. In comparison, we assume the leader knows the reward functions and the main challenge lies in the unknown and leader-controlled transitions. Thus, our setting is different from that in Bai et al. (2021). Furthermore, a more relevant work is (Bucarey et al., 2019b), which establishes the Bellman equation and value iteration algorithm for solving SNE in leader-controlled Markov games. In comparison, we establish modifications of least-squares value iteration that are tailored to online and offline settings.

**Notation** See §B for details.

## 2 PRELIMINARIES

In this section, we introduce the formulation of the general-sum simultaneous-move Markov games, Stackelberg-Nash equilibrium, and the linear structure we use in this paper.

### 2.1 GENERAL-SUM SIMULTANEOUS-MOVE MARKOV GAMES

In this setting, two levels of hierarchy in decision making are considered: one leader $l$ and $N$ followers $\{f_i\}_{i\in[N]}$. Specifically, we define an episodic version of general-sum simultaneous-moves Markov game by the tuple $(\mathcal{S}, \mathcal{A}_l, \mathcal{A}_f = \{\mathcal{A}_{f_i}\}_{i\in[N]}, H, r_l, r_f = \{r_{f_i}\}_{i\in[N]}, \mathcal{P})$, where $\mathcal{S}$ is the state space, $\mathcal{A}_l$ and $\mathcal{A}_f$ are the sets of actions of the leader and the followers respectively, $H$ is the number of steps in each episode, $r_l = \{r_{l,h} : \mathcal{S} \times \mathcal{A}_l \times \mathcal{A}_f \to [-1,1]\}_{h=1}^H$ and $r_{f_i} = \{r_{f_i,h} : \mathcal{S} \times \mathcal{A}_l \times \mathcal{A}_f \to [-1,1]\}_{h=1}^H$ are reward functions of the leader and the followers respectively, and $\mathcal{P} = \{\mathcal{P}_h : \mathcal{S} \times \mathcal{A}_l \times \mathcal{A}_f \times \mathcal{S} \to [0,1]\}_{h=1}^H$ is a collection of transition kernels. Here $\mathcal{A}_l \times \mathcal{A}_f = \mathcal{A}_l \times \mathcal{A}_{f_1} \times \cdots \times \mathcal{A}_{f_N}$. Throughout this paper, we also let $\star$ be some element in $\{l, f_1, \cdots, f_N\}$. Moreover, for any $(h, x, a) \in [H] \times \mathcal{S} \times \mathcal{A}_l$ and $b = \{b_i \in \mathcal{A}_{f_i}\}_{i\in[N]}$, we use the shorthands $r_{\star,h}(x, a, b) = r_{\star,h}(x, a, b_1, \cdots, b_N)$ and $\mathcal{P}_h(\cdot \mid x, a, b) = \mathcal{P}_h(\cdot \mid x, a, b_1, \cdots, b_N)$.

**Policy and Value Function.** A stochastic policy $\pi = \{\pi_h : \mathcal{S} \to \Delta(\mathcal{A}_l)\}_{h=1}^H$ of the leader is a set of probability distributions over actions given the state. Meanwhile, a stochastic joint policy of the followers is defined by $\nu = \{\nu_{f_i}\}_{i\in[N]}$, where $\nu_{f_i} = \{\nu_{f_i,h} : \mathcal{S} \to \Delta(\mathcal{A}_{f_i})\}_{h=1}^H$. We use the notation $\pi_h(a \mid x)$ and $\nu_{f_i,h}(b_i \mid x)$ to denote the probability of taking action $a \in \mathcal{A}_l$ or $b_i \in \mathcal{A}_{f_i}$ for state $x$ at step $h$ under policy $\pi, \nu_{f_i}$ respectively. Throughout this paper, for any $\nu = \{\nu_{f_i}\}_{i\in[N]}$ and $b = \{b_i\}_{i\in[N]}$, we use the shorthand $\nu_h(b \mid x) = \nu_{f_1,h}(b_1 \mid x) \times \cdots \times \nu_{f_N,h}(b_N \mid x)$.

Given policies $(\pi, \nu = \{\nu_{f_i}\}_{i\in[N]})$, the action-value (Q) and state-value (V) functions for the leader and followers are defined by

$$Q_{\star,h}^{\pi,\nu}(x,a,b) = \mathbb{E}_{\pi,\nu,h,x,a,b}\left[\sum_{t=h}^H r_{\star,h}(x_t,a_t,b_t)\right], \quad V_{\star,h}^{\pi,\nu}(x) = \mathbb{E}_{a\sim\pi_h(\cdot\mid x), b\sim\nu_h(\cdot\mid x)}Q_{\star,h}^{\pi,\nu}(x,a,b), \quad (2.1)$$

where the expectation $\mathbb{E}_{\pi,\nu,h,x,a,b}$ is taken over state-action pairs induced by the policies $(\pi, \nu = \{\nu_{f_i}\}_{i\in[N]})$ and the transition probability, when initializing the process with the triplet $(s, a, b = \{b_i\}_{i\in[N]})$ at step $h$. For notational simplicity, when $h, x, a, b$ are clear from the context, we omit $h, x, a, b$ from $\mathbb{E}_{\pi,\nu,h,x,a,b}$. By the definition in (2.1), we have the Bellman equation

$$V_{\star,h}^{\pi,\nu} = \langle Q_{\star,h}^{\pi,\nu}, \pi_h \times \nu_h \rangle_{\mathcal{A}_l \times \mathcal{A}_f}, \quad Q_{\star,h}^{\pi,\nu} = r_{\star,h} + \mathbb{P}_h V_{\star,h+1}^{\pi,\nu}, \quad \forall \star \in \{l, f_1, \cdots, f_N\}, \quad (2.2)$$

where $\pi_h \times \nu_h$ represents $\pi_h \times \nu_{f_1,h} \times \cdots \times \nu_{f_N,h}$. Here $\mathbb{P}_h$ is the operator which is defined by

$$(\mathbb{P}_h f)(x,a,b) = \mathbb{E}[f(x') \mid x' \sim \mathcal{P}_h(x' \mid x, a, b)] \quad (2.3)$$

for any function $f : \mathcal{S} \to \mathbb{R}$ and $(x, a, b) \in \mathcal{S} \times \mathcal{A}_l \times \mathcal{A}_f$.

### 2.2 STACKELBERG-NASH EQUILIBRIUM

Given a leader policy $\pi$, a Nash equilibrium (Nash, 2016) of the followers is a joint policy $\nu^* = \{\nu_{f_i}^*\}_{i\in[N]}$, such that for any $x \in \mathcal{S}$ and $(i, h) \in [N] \times [H]$

$$V_{f_i,h}^{\pi,\nu^*}(x) \geq V_{f_i,h}^{\pi,\nu_{f_i},\nu_{f_{-i}}^*}(x), \quad \forall \nu_{f_i}. \quad (2.4)$$

Here $-i$ represents all indices in $[N]$ except $i$. For each leader policy $\pi$, we denote the set of best-response policies of the followers by $\text{BR}(\pi)$, which is defined by

$$\text{BR}(\pi) = \{\nu = \{\nu_{f_i}\}_{i \in [N]} \,|\, \nu \text{ is the NE of the followers given the leader policy } \pi\}. \quad (2.5)$$

Given the best-response set $\text{BR}(\pi)$, we denote $\nu^*(\pi)$ the worst-case responses, which break ties against favor of the leader [1]. Specifically, we define $\nu^*(\pi)$ by

$$\nu^*(\pi) = \{\nu \in \text{BR}(\pi) \,|\, V_{l,h}^{\pi,\nu}(x) \leq V_{l,h}^{\pi,\nu'}(x), \forall x \in \mathcal{S}, h \in [H], \nu' \in \text{BR}(\pi)\}. \quad (2.6)$$

The Stackelberg-Nash equilibrium for the leader is the "best response to the best response", that is,

$$\text{SNE}_l = \{\pi \,|\, V_{l,h}^{\pi,\nu^*(\pi)}(x) \geq V_{l,h}^{\pi',\nu^*(\pi')}(x), \forall x \in \mathcal{S}, h \in [H], \pi'\} \quad (2.7)$$

A Stackelberg-Nash equilibrium of the general-sum game is a policy pair $(\pi^*, \nu^* = \{\nu_{f_i}^*\}_{i \in [N]})$ such that $\nu^* \in \nu^*(\pi^*)$ and $\pi^* \in \text{SNE}_l$.

Our goal is to find the Stackelberg equilibrium: the leader's optimal strategy, assuming the followers play their best response (Nash equilibrium) to the leader. We study this challenging bilevel optimization problem in both the online setting (Section 3) and the offline setting (Section 4).

## 2.3 Leader-Controller Linear Markov Games

Inspired by the linear MDP studied in Jin et al. (2020b) for the single-agent RL, we study the linear Markov games (Xie et al., 2020), where the transition dynamics are linear in a feature map. Specifically, there exists a feature map $\phi' : \mathcal{S} \times \mathcal{A}_l \times \mathcal{A}_f \to \mathbb{R}^d$ such that

$$\mathcal{P}_h(\cdot \,|\, x, a, b) = \langle \phi'(x, a, b), \mu_h(\cdot) \rangle$$

for any $(x, a, b) \in \mathcal{S} \times \mathcal{A}_l \times \mathcal{A}_f$ and $h \in [H]$. Here $\mu_h = (\mu_h^{(1)}, \mu_h^{(2)}, \cdots, \mu_h^{(d)})$ are $d$ unknown signed measures over $\mathcal{S}$. Moreover, throughout this paper, we focus on the leader-controller game (Filar & Vrieze, 2012; Bucarey et al., 2019a), where the future state only depends on the current state and the leader's action, that is,

$$\mathcal{P}_h(\cdot \,|\, x, a, b) = \mathcal{P}_h(\cdot \,|\, x, a)$$

for any $(x, a, b) \in \mathcal{S} \times \mathcal{A}_l \times \mathcal{A}_f$ and $h \in [H]$. Hence, it is naturally to define leader-controller linear Markov games as follows.

**Assumption 2.1.** Markov game $(\mathcal{S}, \mathcal{A}_l, \mathcal{A}_f = \{\mathcal{A}_{f_i}\}_{i \in [N]}, H, r_l, r_f = \{r_{f_i}\}_{i \in [N]}, \mathcal{P})$ is a leader-controller linear Markov game if there exists a feature map $\phi : \mathcal{S} \times \mathcal{A}_l \to \mathbb{R}^d$ such that

$$\mathcal{P}_h(\cdot \,|\, x, a, b) = \langle \phi(x, a), \mu_h(\cdot) \rangle$$

for any $(x, a, b) \in \mathcal{S} \times \mathcal{A}_l \times \mathcal{A}_f$ and $h \in [H]$. Here $\mu_h = (\mu_h^{(1)}, \mu_h^{(2)}, \cdots, \mu_h^{(d)})$ are $d$ unknown signed measures over $\mathcal{S}$. Without loss of generality, we assume that $\|\mu_h(\mathcal{S})\| \leq \sqrt{d}$ for all $h \in [H]$.

The linear Markov game above is an extension of linear MDP studied in Jin et al. (2020b) for the single-agent RL. Specifically, when the followers play fixed and known policies, the linear Markov games reduce to the linear MDP.

## 3 Main Results for the Online Setting

In this section, we study the online setting, where a central controller controls one leader $l$ and $N$ followers $\{f_i\}_{i \in [N]}$. Our goal is to learn a Stackelberg-Nash equilibrium. In what follows, we formally describe the setup and learning objectives, and then present our algorithm and provide theoretic guarantees.

---

[1]This is also known as pessimistic tie breaking (Conitzer & Sandholm, 2006). We also remark that our subsequent analysis still holds for the optimistic setting (Breton et al., 1988; Bucarey et al., 2019a).

### 3.1 SETUP AND LEARNING OBJECTIVE

We consider the setting where the reward functions $r_l$ and $r_f = \{r_{f_i}\}_{i \in [N]}$ are revealed to the learner before the game. This is reasonable since in practice the reward functions are usually artificially designed. Moreover, we focus on the episodic setting. Specifically, a Markov game is played for $K$ episodes, each of which consists of $H$ timesteps. At the beginning of the $k$-th episode, the leader and followers determine their policies $(\pi^k, \nu^k = \{\nu_{f_i}^k\}_{i \in [N]})$, and a fixed initial state $x_1^k = x_1$ is chosen. Here we assume the fixed initial state just for ease of presentation, and our subsequent results can be generalized to the setting where $x_1^k$ is picked from a fixed distribution. Then the game proceeds as follows. At each step $h \in [H]$, the leader and the followers observe state $x_h^k \in \mathcal{S}$ and pick their own actions $a_h^k \sim \pi_h^k(\cdot \mid x_h^k)$ and $b_h^k = \{b_{i,h}^k \sim \nu_{f_i,h}^k(\cdot \mid x_h^k)\}_{i \in [N]}$. Subsequently, the environment transitions to the next state $x_{h+1}^k \sim \mathcal{P}_h(\cdot \mid x_h^k, a_h^k, b_h^k)$. Each episode terminates after $H$ timesteps.

**Learning Objective.** By the definition in (2.5), given a leader's policy, the best response for the followers is the Nash equilibrium of followers' game induced by this leader's policy. Recall the definition of Nash equilibrium in (2.4), for any policies $(\pi, \nu = \{\nu_{f_i}\}_{i \in [N]})$, it is natural to define the following objective to measure the suboptimality of $\nu_{f_i}$:

$$\text{SubOpt}_{f_i}(x) = V_{f_i,1}^{\pi,\nu^*(\pi)}(x) - V_{f_i,1}^{\pi,\nu_{f_i},\nu_{-i}^*(\pi)}(x).$$

Meanwhile, we evaluate the performance of the leader's policy $\pi$ by the following suboptimality gap:

$$\text{SubOpt}_l(x) = V_{l,1}^{\pi^*,\nu^*}(x) - V_{l,1}^{\pi,\nu^*(\pi)}(x).$$

Putting these two suboptimality gaps together, we formally define the regret as follows.

**Definition 3.1** (Regret). Let $(\pi^k, \nu^k = \{\nu_{f_i}^k\}_{i \in [N]})$ denote the policies executed by the algorithm in the $k$-th episode. After a total of $K$ episodes, the regret is defined as

$$\text{Regret}(K) = \underbrace{\sum_{k=1}^{K} V_{l,1}^{\pi^*,\nu^*}(x_1^k) - V_{l,1}^{\pi^k,\nu^*(\pi^k)}(x_1^k)}_{\text{Regret}_l(K)} + \underbrace{\sum_{i=1}^{N}\sum_{k=1}^{K} V_{f_i,1}^{\pi^k,\nu^*(\pi^k)}(x_1^k) - V_{f_i,1}^{\pi^k,\nu_{f_i}^k,\nu_{f_{-i}}^*(\pi^k)}(x_1^k)}_{\text{Regret}_f(K)}.$$

$$(3.1)$$

The goal is to design algorithms with regret that is sublinear in $K$, and polynomial in $d, H$. Here $K$ is the number of episodes, $d$ is the dimension of the feature map $\phi$, and $H$ is the episode horizon.

### 3.2 ALGORITHM

We now present our algorithm, Optimistic Value Iteration to Find Stackelberg-Nash Equilibrium (OVI-SNE), which is given in Algorithm 1.

At a high level, in each episode, our algorithm first construct the policies for all players through backward induction with respect to the timestep $h$ (line 4-11), and then execute the policies to play the game (line 12-16).

In detail, at $h$-th step of $k$-th episode, OVI-SNE estimates leader's Q-function based on the $(k-1)$ historical trajectories. Inspired by previous optimistic least square value iteration (LSVI) algorithms (Jin et al., 2020b), for any $h \in [H]$, we estimate the linear coefficients by solving the following ridge regression problem:

$$w_h^k \leftarrow \underset{w \in \mathbb{R}^d}{\text{argmin}} \sum_{\tau=1}^{k-1} [V_{h+1}^k(x_{h+1}^\tau) - \phi(x_h^\tau, a_h^\tau)^\top w]^2 + \|w\|^2,$$

$$\text{where } V_{h+1}^k(\cdot) = \langle Q_{h+1}^k(\cdot, \cdot, \cdot), \pi_{h+1}^k(\cdot \mid \cdot) \times \nu_{h+1}^k(\cdot \mid \cdot) \rangle_{\mathcal{A}_l \times \mathcal{A}_f}.$$

$$(3.2)$$

By solving the ridge regression problem in (3.2), we have

$$w_h^k = (\Lambda_h^k)^{-1} \Big( \sum_{\tau=1}^{k-1} \phi(x_h^\tau, a_h^\tau) \cdot V_{h+1}^k(x_{h+1}^\tau) \Big),$$

$$\text{where } \Lambda_h^k = \sum_{\tau=1}^{k-1} \phi(x_h^\tau, a_h^\tau)\phi(x_h^\tau, a_h^\tau)^\top + I.$$

$$(3.3)$$

To encourage exploration, we additionally adds a bonus function to estimate the leader's Q-function:

$$Q_h^k(\cdot,\cdot,\cdot) \leftarrow r_{l,h}(\cdot,\cdot,\cdot) + \Pi_{H-h}\{\phi(\cdot,\cdot)^\top w_h^k + \Gamma_h^k(\cdot,\cdot)\},$$
$$\text{where } \Gamma_h^k(\cdot,\cdot) = \beta \cdot \sqrt{\phi(\cdot,\cdot)^\top (\Lambda_h^k)^{-1}\phi(\cdot,\cdot)}. \tag{3.4}$$

Here $\Gamma_h^k : \mathcal{S} \times \mathcal{A}_l \to \mathbb{R}$ is a bonus function and $\beta > 0$ is a parameter which will be specified later. This form of bonus function is common in the literature of linear bandits (Lattimore & Szepesvári, 2020) and linear MDPs (Jin et al., 2020b).

Then, we construct policies for the leader and followers by the subroutine $\epsilon$-SNE (Algorithm 2). Specifically, let $\mathcal{Q}_h^k$ be the class of functions $Q : \mathcal{S} \times \mathcal{A}_l \times \mathcal{A}_f \to \mathbb{R}$ that takes form

$$Q(\cdot,\cdot,\cdot) = r_{l,h}(\cdot,\cdot,\cdot) + \Pi_{H-h}\{\phi(\cdot,\cdot)^\top w + \beta \cdot (\phi(\cdot,\cdot)^\top \Lambda^{-1}\phi(\cdot,\cdot))^{1/2}\}, \tag{3.5}$$

where the parameters $(w, \Lambda) \in \mathbb{R}^d \times \mathbb{R}^{d \times d}$ satisfy $\|w\| \leq H\sqrt{dk}$ and $\lambda_{\min}(\Lambda) \geq 1$. Moreover, let $\mathcal{Q}_{h,\epsilon}^k$ be a fixed $\epsilon$-covering of $\mathcal{Q}_h^k$ with respect to the $\ell_\infty$ norm. By Lemma C.10, we have $Q_h^k \in \mathcal{Q}_h^k$, which allows us to pick a $\widetilde{Q} \in \mathcal{Q}_{h,\epsilon}^k$ such that $\|\widetilde{Q} - Q_h^k\|_\infty \leq \epsilon$ and calculate policies by

$$(\pi_h^k(\cdot\,|\,x), \{\nu_{f_i,h}^k(\cdot\,|\,x)\}_{i\in[N]}) \leftarrow \text{SNE}(\widetilde{Q}(x,\cdot,\cdot), \{r_{f_i,h}(x,\cdot,\cdot)\}_{i\in[N]}), \forall x. \tag{3.6}$$

When there is only one follower, such a problem can be transformed to a linear programming (LP) problem (Conitzer & Sandholm, 2006; Von Stengel & Zamir, 2010), and thus can be solved efficiently. For the multi-follower case, however, solving such a matrix game in general is hard (Conitzer & Sandholm, 2006; Basilico et al., 2017a;b; Coniglio et al., 2020). Given this computational hardness, we focus on the sample complexity and explicitly assume access to the following computational oracle:

**Assumption 3.2.** We assume access to an oracle that implements Line 3 of Algorithm 2 when there are multiple followers (i.e., $N \geq 2$).

Now we explain the motivation for using the subroutine $\epsilon$-SNE to construct policies instead of solving the matrix games with payoff matrices $(Q_h^k(x,\cdot,\cdot), \{r_{f_i,h}(x,\cdot,\cdot)\}_{i\in[N]})$ directly. By the definition of $Q_h^k$ in (3.4), we know $Q_h^k$ relies on the previous data via the estimated value function $V_{h+1}^k$ and feature maps $\{\phi(x_h^\tau, a_h^\tau, b_h^\tau)\}_{\tau=1}^{k-1}$. Similar to the analysis for linear MDPs (Jin et al., 2020b), we need to use a covering argument to establish uniform concentration bounds for all value $V_{h+1}^k$. Jin et al. (2020b) directly constructs an $\epsilon$-net for the value functions and establishes a polynomial log-covering number for this $\epsilon$-net. This analysis, however, relies on that the policies executed by the players are greedy (deterministic), which is not valid for our setting. To overcome this technical issue, we construct an $\epsilon$-net for Q-functions and solve an approximate matrix game. Fortunately, by choosing a small enough $\epsilon$, we can handle the errors caused by this approximation. See §C for more details. Moreover, as shown in Xie et al. (2020), this subroutine can be implemented efficiently without explicitly computing the exponentially large $\epsilon$-net.

Finally, the leader and the followers play the game according to the obtained policies.

### 3.3 THEORETICAL RESULTS

Our main theoretical result is the following bound on the regret incurred by Algorithm 1. Recall that the regret is defined in Definition 3.1 and $T = KH$ is the total number of timesteps.

**Theorem 3.3.** Under Assumptions 2.1 and 3.2, there exists an absolute constant $C > 0$ such that, for any fixed $p \in (0, 1)$, by setting $\beta = C \cdot dH\sqrt{\iota}$ with $\iota = \log(2dT/p)$ in Line 7 of Algorithm 1 and $\epsilon = \frac{1}{KH}$ in Algorithm 2, then with probability at least $1 - p$, the regret incurred by OVI-SNE satisfies that

$$\text{Regret}(K) \leq \mathcal{O}(\sqrt{d^3 H^3 T \iota^2}).$$

*Proof.* See §C for a detailed proof. □

**Learning Stackelberg Equilibria.** When there is only one follower, Stackelberg-Nash equilibrium reduces to the Stackelberg equilibrium (Simaan & Cruz, 1973; Conitzer & Sandholm, 2006; Bai et al., 2021). Thus, we partly answer the open problem in Bai et al. (2021) on how to learn Stackelberg equilibria in (leader-controller) Markov games.

---

**Algorithm 1** Optimistic Value Iteration to Find Stackelberg-Nash Equilibria

---

1: Initialize $V_{l,H+1}(\cdot) = V_{f,H+1}(\cdot) = 0$.
2: **for** $k = 1, 2, \cdots, K$ **do**
3:     Receive initial state $x_1^k$.
4:     **for** step $h = H, H-1, \cdots, 1$ **do**
5:         $\Lambda_h^k \leftarrow \sum_{\tau=1}^{k-1} \phi(x_h^\tau, a_h^\tau)\phi(x_h^\tau, a_h^\tau)^\top + I$.
6:         $w_h^k \leftarrow (\Lambda_h^k)^{-1} \sum_{\tau=1}^{k-1} \phi(x_h^\tau, a_h^\tau) \cdot V_{h+1}^k(x_{h+1}^\tau)$.
7:         $\Gamma_h^k(\cdot, \cdot) \leftarrow \beta \cdot (\phi(\cdot, \cdot)^\top (\Lambda_h^k)^{-1} \phi(\cdot, \cdot))^{1/2}$.
8:         $Q_h^k(\cdot, \cdot, \cdot) \leftarrow r_{l,h}(\cdot, \cdot, \cdot) + \Pi_{H-h}\{\phi(\cdot, \cdot)^\top w_h^k + \Gamma_h^k(\cdot, \cdot)\}$.
9:         $(\pi_h^k(\cdot \,|\, x), \{\nu_{f_i,h}^k(\cdot \,|\, x)\}_{i \in [N]}) \leftarrow \epsilon\text{-SNE}(Q_h^k(x, \cdot, \cdot), \{r_{f_i,h}(x, \cdot, \cdot)\}_{i \in [N]}), \; \forall x.$ (Alg. 2)
10:        $V_h^k(x) \leftarrow \mathbb{E}_{a \sim \pi_h^k(\cdot \,|\, x), b_1 \sim \nu_{f_1,h}^k(\cdot \,|\, x), \cdots, b_N \sim \nu_{f_N,h}^k(\cdot \,|\, x)} Q_h^k(x, a, b_1, \cdots, b_N), \; \forall x.$
11:    **end for**
12:    **for** $h = 1, 2, \cdot, H$ **do**
13:        Sample $a_h^k \sim \pi_h^k(\cdot \,|\, x_h^k), b_{1,h}^k \sim \nu_{f_1,h}^k(\cdot \,|\, x_h^k), \cdots, b_{N,h}^k \sim \nu_{f_N,h}^k(\cdot \,|\, x_h^k)$.
14:        Leader takes action $a_h^k$; Followers take actions $b_h^k = \{b_{i,h}^k\}_{i \in [N]}$.
15:        Observe next state $x_{h+1}^k$.
16:    **end for**
17: **end for**

---

**Algorithm 2** $\epsilon$-SNE

---

1: **Input:** $Q_h^k, x$, and parameter $\epsilon$.
2: Select $\widetilde{Q}$ from $\mathcal{Q}_{h,\epsilon}^k$ satisfying $\|\widetilde{Q} - Q_h^k\|_\infty \le \epsilon$.
3: For the input state $x$, let $(\pi_h^k(\cdot \,|\, x), \{\nu_{f_i,h}^k(\cdot \,|\, x)\}_{i \in [N]})$ be the Stackelberg-Nash equilibrium for the matrix game with payoff matrices $(\widetilde{Q}(x, \cdot, \cdot), \{r_{f_i,h}(x, \cdot, \cdot)\}_{i \in [N]})$.
4: **Output:** $(\pi_h^k(\cdot \,|\, x), \{\nu_{f_i,h}^k(\cdot \,|\, x)\}_{i \in [N]})$.

---

**Optimality of the Bound.** Assuming that the action of the follower won't affect the transition kernel and reward function, the linear Markov games reduces to the linear MDP (Jin et al., 2020b). Meanwhile, the lower bound established in Azar et al. (2017); Jin et al. (2018) for tabular MDPs and the lower bound established in Lattimore & Szepesvári (2020) for linear bandits directly imply a lower bound $\Omega(dH\sqrt{T})$ for the linear MDPs, which further yields a lower bound $\Omega(dH\sqrt{T})$ for our setting. Ignoring the logarithmic factors, there is only a gap of $\sqrt{dH}$ between this lower bound and our upper bound. We also point out that, by using the "Bernstein-type" bonus (Azar et al., 2017; Jin et al., 2018; Zhou et al., 2020), we can improve our upper bound by a factor of $\sqrt{H}$. Here we don't apply this technique for the clarity of the analysis.

**Misspecification.** For ease of presentation, we assume the Markov games are leader-controller in Assumption 2.1. When the transitions do not ideally satisfy the leader-controller assumption, we can potentially consider cases that transitions satisfy, for instance, $\|\mathcal{P}_h(\cdot \,|\, x, a, b) - \mathcal{P}_h(\cdot \,|\, x, a)\|_\infty \le \varrho$ for any $(h, x, a, b) \in [H] \times \mathcal{S} \times \mathcal{A}_l \times \mathcal{A}_f$, Here $\varrho$ is the misspecification error. We can still follow the above method to tackle the misspecified cases. However, because of the misspecification error cumulated during $T$ steps, an extra term $\mathcal{O}(\varrho T)$ will appear in the final result. In particular, When $\varrho$ is small, that is the Markov games have approximately leader-controller transitions, the extra term $\mathcal{O}(\varrho T)$ should be small, which further indicates that we can find SNEs efficiently in some misspecified general-sum Markov games.

**Unknown Reward Setting.** At a high level, we first conduct a reward-free exploration algorithm (Algorithm 4 in §D), a variant of Reward-Free RL-Explore algorithm in Jin et al. (2020a), to obtain estimated reward functions $\{\widehat{r}_l, \widehat{r}_{f_1}, \cdots \widehat{r}_{f_N}\}$. As asserted before, we can use Algorithm 1, to find the SNE with respect to the *known* estimated reward functions $\{\widehat{r}_l, \widehat{r}_{f_1}, \cdots \widehat{r}_{f_N}\}$. Hence, we can obtain the approximate SNE if the value functions of estimated value functions are good approximation of the true value functions. See §E for more details.

# 4 MAIN RESULTS FOR THE OFFLINE SETTING

In this section, we study the offline setting, where the central controller aims to find a Stackelberg-Nash equilibrium by an offline dataset. Below we describe the setup and learning objective, followed by our algorithm and theoretical results.

## 4.1 SETUP AND LEARNING OBJECTIVE

We study the offline setting, where the learner has access to the reward functions $(r_l, r_f = \{r_{f_i}\}_{i=1}^N)$ and a dataset $\mathcal{D} = \{(x_h^\tau, a_h^\tau, b_h^\tau = \{b_{i,h}^\tau\}_{i=1}^N)\}_{\tau,h=1}^{K,H}$, which is collected a priori by some experimenter. Then we make a minimal assumption for the offline dataset.

**Assumption 4.1** (Compliance of Dataset). We assume that the dataset $\mathcal{D}$ is compliant with the underlying Markov game $(\mathcal{S}, \mathcal{A}_l, \mathcal{A}_f, H, r_l, r_f, \mathcal{P})$, that is, for any $x' \in \mathcal{S}$ at step $h \in [H]$ of each trajectory $\tau \in [K]$,

$$P_\mathcal{D}(x_{h+1}^\tau = x' \mid \{x_h^j, a_h^j, b_h^j, x_{h+1}^j\}_{j=1}^{\tau-1} \cup \{x_h^\tau, a_h^\tau, b_h^\tau\}) = P(x_{h+1} = x' \mid x_h = x_h^\tau, a_h = a_h^\tau).$$

Here the probability on the left-hand side is with respect to the joint distribution over dataset $\mathcal{D}$ and the probability on the right-hand side is with respect to the underlying Markov game.

Assumption 4.1 is adopted from Jin et al. (2020c), which indicates the Markov property of the dataset $\mathcal{D}$ and that $x_{h+1}^\tau$ is generated by the underlying Markov game conditioned on $(x_h^\tau, a_h^\tau, b_h^\tau)$. As a special case, Assumption 4.1 holds when the experimenter follows fixed behavior policies. More generally, Assumption 4.1 allows the experimenter to choose actions $a_h^\tau$ and $b_h^\tau$ arbitrarily, even in an adaptive or adversarial manner. In particular, we can assume that $a_h^\tau$ and $b_h^\tau$ are interdependent across each trajectory $\tau \in [K]$. For instance, the experimenter can sequentially improve the behavior policy using any online algorithm for Markov games.

**Learning Objective.** Similar to the online setting, we define the following performance metric

$$\text{SubOpt}(\pi, \nu, x) = \underbrace{V_{l,1}^{\pi^*, \nu^*}(x) - V_{l,1}^{\pi, \nu^*(\pi)}(x)}_{\text{SubOpt}_l} + \underbrace{\sum_{i=1}^N [V_{f_i,1}^{\pi, \nu^*(\pi)}(x) - V_{f_i,1}^{\pi, \nu_{f_i}, \nu_{f_{-i}}^*(\pi)}(x)]}_{\text{SubOpt}_f}, \quad (4.1)$$

which evaluates the suboptimality of policies $(\pi, \nu = \{\nu_{f_i}\}_{i=1}^N)$ given the initial state $x \in \mathcal{S}$.

## 4.2 ALGORITHM AND THEORETICAL RESULTS

As is known to us, the key challenge of online setting is the the tradeoff between exploration and exploration. In the online setting. by following the "optimism in the face of uncertainty" principle (Sutton & Barto, 2018), we use bonus functions to incentivize exploration and thus achieve sample-efficient. This intrinsic challenge of online setting disappears in the offline setting because we do not need exploration any more. But another challenge arises: we only have access to the limited data. To tackle this challenge, we need add some penalty functions to achieve robustness against the uncertainty due to the finite data. This is also known as pessimism (Yu et al., 2020; Jin et al., 2020c; Liu et al., 2020b; Buckman et al., 2020; Kidambi et al., 2020; Kumar et al., 2020; Rashidinejad et al., 2021). Here we simply flip the sign of bonus functions defined in (3.4) to serve as penalty functions. See Algorithm 3 for details.

Suppose that $(\widehat{\pi}, \widehat{\nu})$ are the output policies of Algorithm 3. Then we evaluate the performance of $(\widehat{\pi}, \widehat{\nu})$ by establishing an upper bound for the optimality gap defined in (4.1).

**Theorem 4.2.** Under Assumptions 2.1, 3.2, and 4.1, there exists an absolute constant $C > 0$ such that, for any fixed $p \in (0, 1)$, by setting $\beta' = C \cdot dH\sqrt{\log(2dHK/p)}$ in Line 6 of Algorithm 3 and $\epsilon = \frac{d}{KH}$ in Algorithm 2, then with probability at least $1 - p$, we have

$$\text{SubOpt}(\widehat{\pi}, \widehat{\nu}, x) \le 3\beta' \sum_{h=1}^H \mathbb{E}_{\pi^*, x}\big[\big(\phi(s_h, a_h)^\top (\Lambda_h)^{-1} \phi(s_h, a_h)\big)^{1/2}\big], \quad (4.2)$$

where $\mathbb{E}_{\pi^*, x}$ is taken with respect to the trajectory incurred by $\pi^*$ in the underlying leader-controller Markov game when initializing the progress at $x$. Here $\Lambda_h$ is defined in Line 4 of Algorithm 3.

*Proof.* See §F for a detailed proof. $\qquad\square$

---

**Algorithm 3** Pessimistic Value Iteration to Find Stackelberg-Nash Equilibria

1: **Input:** $\mathcal{D} = \{x_h^\tau, a_h^\tau, b_h^\tau = \{b_{i,h}^\tau\}_{i\in[N]}\}_{\tau,h=1}^{K,H}$ and reward functions $\{r_l, r_f = \{r_{f_i}\}_{i\in[N]}\}$.
2: Initialize $\widehat{V}_{H+1}(\cdot) = 0$.
3: **for** step $h = H, H-1, \cdots, 1$ **do**
4: $\quad \Lambda_h \leftarrow \sum_{\tau=1}^K \phi(x_h^\tau, a_h^\tau)\phi(x_h^\tau, a_h^\tau)^\top + I$.
5: $\quad w_h \leftarrow (\Lambda_h)^{-1}\sum_{\tau=1}^K \phi(x_h^\tau, a_h^\tau) \cdot \widehat{V}_{h+1}(x_{h+1}^\tau)$.
6: $\quad \Gamma_h(\cdot,\cdot) \leftarrow \beta' \cdot (\phi(\cdot,\cdot)^\top (\Lambda_h)^{-1}\phi(\cdot,\cdot))^{1/2}$.
7: $\quad \widehat{Q}_h(\cdot,\cdot,\cdot) \leftarrow r_{l,h}(\cdot,\cdot,\cdot) + \Pi_{H-h}\{\phi(\cdot,\cdot)^\top w_h - \Gamma_h(\cdot,\cdot)\}$.
8: $\quad (\widehat{\pi}_h(\cdot\,|\,x), \{\widehat{\nu}_{f_i,h}(\cdot\,|\,x)\}_{i\in[N]}) \leftarrow \epsilon\text{-SNE}(\widehat{Q}_h(x,\cdot,\cdot), \{r_{f_i,h}(x,\cdot,\cdot)\}_{i\in[N]}), \forall x.$ (Alg. 2)
9: $\quad \widehat{V}_h(x) \leftarrow \mathbb{E}_{a\sim\widehat{\pi}_h(\cdot\,|\,x), b_1\sim\widehat{\nu}_{f_1,h}(\cdot\,|\,x),\cdots,b_N\sim\widehat{\nu}_{f_N,h}(\cdot\,|\,x)}\widehat{Q}_h(x,a,b_1,\cdots,b_N), \forall x.$
10: **end for**
11: **Output:** $(\widehat{\pi} = \{\widehat{\pi}_h\}_{h=1}^H, \widehat{\nu} = \{\widehat{\nu}_{f_i} = \{\nu_{f_i,h}\}_{h=1}^H\}_{i=1}^N)$.

---

**Minimal Assumption Requirement:** Theorem 4.2 only relies on the compliance of the dataset with linear Markov games. Compared with existing literature on offline RL (Bertsekas & Tsitsiklis, 1996; Antos et al., 2007; 2008; Munos & Szepesvári, 2008; Farahmand et al., 2010; 2016; Scherrer et al., 2015; Liu et al., 2018; Chen & Jiang, 2019; Fan et al., 2020; Xie & Jiang, 2020), we impose no restrictions on the coverage of the dataset. Meanwhile, we need no assumption on the affinity between $(\widehat{\pi}, \widehat{\nu})$ and the behavior policies that induce the dataset, which is often employed as a regularizer (Fujimoto et al., 2019; Laroche et al., 2019; Jaques et al., 2019; Wu et al., 2019; Kumar et al., 2019; Wang et al., 2020; Siegel et al., 2020; Nair et al., 2020; Liu et al., 2020b).

**Dataset with Sufficient Coverage:** In what follows, we specialize Theorem 4.2 to the setting where we assume the dataset with good "coverage". Note that $\Lambda_h$ is determined by the offline dataset $\mathcal{D}$ and acts as a fixed matrix in the expectation, that is, the expectation in (4.2) is only taken with the trajectory induced by $\pi^*$. As proofed in the following theorem, when the trajectory induced by $\pi^*$ is "covered" by the dataset $\mathcal{D}$ sufficiently well, we can establish that the suboptimality incurred by Algorithm 3 diminishes at rate of $\widetilde{\mathcal{O}}(1/\sqrt{K})$.

**Corollary 4.3.** Suppose it holds with probability at least $1 - p/2$ that

$$\Lambda_h \succeq I + c \cdot K \cdot \mathbb{E}_{\pi^*,x}[\phi(s_h,a_h)\phi(s_h,a_h)^\top]$$

for all $(x, h) \in \mathcal{S} \times [H]$. Here $c > 0$ is an absolute constant and $\mathbb{E}_{\pi^*,x}$ is taken with respect to the trajectory incurred by $\pi^*$ in the underlying leader-controller Markov game when initializing the progress at $x$. Under Assumptions 2.1, 3.2 and 4.1, there exists an absolute constant $C > 0$ such that, for any fixed $p \in (0,1)$, by setting $\beta' = C \cdot dH\sqrt{\log(4dHK/p)}$ in Line 6 of Algorithm 3 and $\epsilon = \frac{d}{KH}$ in Algorithm 2, then it holds with probability at least $1 - p$ that

$$\text{SubOpt}(\widehat{\pi}, \widehat{\nu}, x) \leq \bar{C} \cdot d^{3/2}H^2\sqrt{\log(4dHK/p)/K}$$

for all $x \in \mathcal{S}$. Here $\bar{C}$ is another absolute constant that only depends on $c$ and $C$.

*Proof.* See §G for a detailed proof. $\qquad\square$

Note that, unlike the previous literature (Antos et al., 2007; Munos & Szepesvári, 2008; Farahmand et al., 2010; 2016; Scherrer et al., 2015; Liu et al., 2018; Chen & Jiang, 2019; Fan et al., 2020; Xie & Jiang, 2020) which relies on the "uniform coverage" assumption, Corollary 4.3 only assumes that the dataset has a good coverage of the trajectory incurred by the policy $\pi^*$.

**Optimality of the Bound:** Assuming the dummy followers, that is, the actions taken by the followers won't affect the reward functions and transition kernels, the Markov games reduces to the linear MDP (Jin et al., 2020b). Together with the information-theoretic lower bound $\Omega(\sum_{h=1}^H \mathbb{E}_{\pi^*,x}[(\phi(s_h,a_h)^\top(\Lambda_h)^{-1}\phi(s_h,a_h))^{1/2}])$ established in Jin et al. (2020c) for linear MDPs, we immediately obtain the same lower bound for our setting. In particular, our upper bound established in Theorem 4.2 matches this lower bound up to $\beta'$ and absolute constants and thus implies that our algorithm is nearly minimax optimal.

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
