# OpenReview forum: "Can Reinforcement Learning Efficiently Find Stackelberg-Nash Equilibria in General-Sum Markov Games?"
_ICLR.cc/2022/Conference — ICLR 2022 Submitted_

### Official Review · Reviewer_cJRF · 2021-10-25

**Correctness:** 4
**Technical Novelty And Significance:** 2
**Empirical Novelty And Significance:** Not applicable
**Recommendation:** 5
**Confidence:** 4

**Main Review:**

Strength: The paper provides a near-optimal and, to my knowledge, novel regret bound for learning Stackelberg equilibria in the chosen setting.

Weaknesses: The setting is not very well motivated. The authors stipulate very significant assumptions on the problem (the existence of a normal-form SNE oracle, and the followers not affecting the state transitions) without justifying them. Why should we care about this very specific setting? Are there interesting problems that fall into it?

Also, the above two assumptions together seem to immediately sidestep everything difficult and interesting about Stackelberg equilibria and multiplayer games. Indeed, the followers' strategy at each time step is now a function of the leader's strategy (because the followers cannot control the transitions), and that function is given by an oracle (by assumption). As such, the problem is reduced immediately to a (single-player) MDP.

In some sense, given the above, what is really going on is that the authors have developed algorithms for solving a class of MDPs perhaps more general than linear MDPs. If that class of MDPs can be isolated and is significant or interesting in its own right, perhaps it would be a good idea to frame the paper in that light instead. I also wonder if directly framing the problem as an MDP results in a simplification of the algorithms or analyses via known tools for MDPs.

**Summary Of The Paper:**

The paper studies the problem of computing Stackelberg-Nash equilibria in a certain restricted family of Markov games, in which the followers do not control the transitions. Assuming an oracle that solves each stage game, the authors develop an algorithm for solving for Stackelberg-Nash equilibria in the overall game.

**Summary Of The Review:**

A perhaps interesting theoretical result, but poorly motivated and very narrow.

---

> ### Author Response · Authors · 2021-11-19
> **To Reviewer cJRF**
>
> Q1: The setting is not very well motivated. The authors stipulate very significant assumptions on the problem (the existence of a normal-form SNE oracle, and the followers not affecting the state transitions) without justifying them. Why should we care about this very specific setting? Are there interesting problems that fall into it?
>
> A1:  We point out that, when there is only one follower, the normal form game can be calculated by linear programming (LP). Finding the Stackelberg equilibrium in the general-sum Markov games remains open before our work. When there are multiple followers, calculating the SNE for the normal-form game is generally hard, which is left as the future work. We also want to emphasize that we mainly focus on establishing $\sqrt{T}$ regret for general-sum Markov games, which remain open before our work.
> As stated in the introduction, this problem has many applications in  (1) Mechanism design (Conitzer & Sandholm, 2002; Roughgarden, 2004; Garg & Narahari, 2005; Kang & Wu, 2014): The leader imposes a price on each unit of bandwidth providing to each downloader. Then, the followers determine their optimal download bandwidths to maximize their individual utilities based on the assigned bandwidth price. (2) Security games (Tambe, 2011; Korzhyk et al., 2011): In this setting, a defender (leader) commits to a randomized deployment of security resources, and an attacker (follower) bestresponds by attacking a target that maximizes his utility. (3) Incentive design (Zheng et al., 1984; Ratliff et al., 2014; Chen et al., 2016; Ratliff & Fiez, 2020): There is a coordinator (planner/leader) and $n$ non–cooperative strategic agents (followers) each providing response. The followers play according to a Nash equilibrium strategy, and the goal of the leader is to design a mechanism to coordinate the followers by incentivizing them to take actions that ultimately lead to the minimization of objective functions. (4) We also give a simplified example in the introduction, which consists of a government and a group of companies. When the transition is determined by the current state and the action of government, this is a leader-controller general sum game.
>
> Q2: Also, the above two assumptions together seem to immediately sidestep everything difficult and interesting about Stackelberg equilibria and multiplayer games. Indeed, the followers' strategy at each time step is now a function of the leader's strategy (because the followers cannot control the transitions), and that function is given by an oracle (by assumption). As such, the problem is reduced immediately to a (single-player) MDP.
>
> A2: We want to emphasize that (1) the Stackelberg game is a bilevel optimization problem, and (2) the rewards depend on all players’ actions, and thus leader’s policy should take the followers’ best response into consideration when improving her policy own policy. Hence our setting is much harder than the single-agent MDP. To tackle these challenges, we propose a novel algorithm that adopts the optimistic estimation of leader’s value function for exploration and solves the bilevel optimization problem for policy optimization. Markov games also have some technical challenges even with the leader-controller assumption. For example, we need to use the $\epsilon$-SNE subroutine due to some technical issues. See the paragraph below Assumption 3.2 for details.
>
> “the problem is reduced immediately to a (single-player) MDP” is wrong. Because the best response function is a highly nonlinear function of the leader’s strategy. So leader‘s problem is not a standard MDP as the total return is no longer a linear function of the visitation measure.

---

> > ### Comment · Reviewer_9LtG · 2021-11-23
> > **Question**
> >
> > So you claim that you are the first that compute efficiently Stackelberg Equilibria in GeneralSum MDPs?

---

> > > ### Author Response · Authors · 2021-11-23
> > > **Response**
> > >
> > > We would like to thank the reviewer for the question.
> > >
> > > Our work is the first that proposes a provably **sample efficient** RL algorithm for learning Stackelberg equilibria in general-sum Markov games. Here, being sample efficient means that to learn an $\epsilon$-optimal policy for the leader, the number of trajectories needed is $\textrm{poly}(d)\cdot \textrm{poly}(H) \cdot \epsilon^{-2}$, which is equivalent to having a $\textrm{poly}(d) \cdot  \textrm{poly}(H)\cdot \sqrt{T}$ regret. Here $d$ is the dimension of the feature mapping and $H$ is the length of the episode.
> > >
> > > Furthermore, in terms of computational efficiency, our method is **oracle efficient** in the sense that the number of queries to the _best response oracle_ of the followers is $\textrm{poly}(H, T)$. Moreover, when there is only _one_ myopic follower, our algorithm is computationally efficient. As for memory efficiency, since our algorithm only needs to store the features and parameters, the memory complexity is also $\text{poly}(d, H, T)$.
> > >
> > > Furthermore, we would like to highlight that our work first tackles the problem of learning Stackelberg equilibria in general-sum Markov games in the *online setting*. Previous works either assume the _transition model is known_, or only consider the _static game_. While our work considers the Markov game setting with state transitions, without the knowledge of the underlying model --- the leader needs to design a proper **exploration** method in order to learn the optimal policy efficiently from data.
> > >
> > > Finally, in terms of the literature on *online learning in RL*, almost all the existing works focus on either *MDP* or *zero-sum Markov games*. In these two problems, the optimization problem is either a *direct maximization* or a *minimax optimization*. Whereas our Stackelberg equilibrium involves bilevel optimization and thus these works are not applicable. Novel exploration schemes are needed for the leader which utilizes such a bilevel structure and considers the best-response behavior of the followers when doing exploration. In fact, our work proves that, combining (i) optimism in the face of uncertainty and (ii) solving a bilevel optimization based on the optimistic value function leads to a provably sample efficient algorithm for the leader.

---

> > ### Comment · Reviewer_cJRF · 2021-11-24
> > **Response to response**
> >
> > I have no doubt as to the general importance of Stackelberg games. However, of the examples you have given, none are obviously problems of the narrow type solved in the paper. In particular, as far as I can tell, most of the examples in (1-3) above are not sequential at all, and those that are sequential are not naturally leader-controlled; thus, the paper's techniques apply to none of them. Example (4) seems rather contrived, in the sense that I have no reason to believe that such a setting could be reasonably treated as leader- (government-) controlled. Could the authors provide some interesting examples (e.g., with citations to other papers) of specifically *leader-controlled, sequential* Stackelberg games, to which the algorithms stated in the paper could apply?

---

> > > ### Author Response · Authors · 2021-11-24
> > > **Examples**
> > >
> > > Before giving examples, we want to clarify that our results still hold when the followers are myopic (the transitions can be determined by all the players). Here myopic followers mean that the followers only maximize the rewards in current steps.
> > >
> > > Finding the Stackelberg problem is generally complicated. To make the problem tractable, various simplifications of the dynamic Stackelberg equilibrium are considered. One of them is a model in which the followers are myopic.
> > >
> > > [1] studies myopic‑follower Stackelberg equilibria in the dynamic market model. They also explain the myopic assumptions in the following two scenarios. (i) The leader is an established firm and the followers are entrants. The followers are myopic because they are not sure whether the firm is going to exist in the future. (ii) Leader is the one who dictates prices and the follower just does not know the pricing rules of the leader and is thus myopic. This also explains why our previous Example (4) is leader- (government-) controlled.
> > >
> > > More examples of marketing and supply chain management can be found in [2].
> > >
> > > From the theoretical side, [3] also studies Stackelberg equilibrium with myopic followers.	However, they assume the known transitions and thus circumvent the challenge of exploration.
> > >
> > > We hope our response clarifies your concerns and you will consider raising your score. Thank you!
> > >
> > > [1] Katarzyna Kańska, Agnieszka Wiszniewska‑Matyszkiel. Dynamic Stackelberg duopoly with sticky prices and a myopic follower. Operational Research, 2021.
> > >
> > > [2] Tao Li, Suresh P Sethi. A review of dynamic Stackelberg game models. Discrete & Continuous Dynamical Systems-B, 2017.
> > >
> > > [3] Víctor Bucarey, Eugenio Della Vecchia, Alain Jean-Marie, Fernando Ordóñez. 	Stationary Strong Stackelberg Equilibrium in Discounted Stochastic Games.

---

> > > > ### Comment · Reviewer_cJRF · 2021-11-25
> > > > **Thank you.**
> > > >
> > > > Awesome. That addresses a major concern of mine, and in light of it I raise my score.
> > > >
> > > > I would recommend the authors to make a major revision to the paper in which they explicitly move to discussing the myopic-follower setting instead of the leader-controlled setting. This would likely entail rewriting large sections of the paper. I would also recommend that the authors include the above discussion and citations about examples involving myopic followers instead of the current third paragraph of the introduction, which, as we discussed, seems to only give examples of games not captured by this paper.
> > > >
> > > > Relatively minor note: I'd encourage a different title that explicitly mentions the myopic-follower setting; the current title feels, in my opinion, too bold given the paper content. (Indeed, the answer to the question in the current title seems to be "no, but we can do this special case".)
> > > >
> > > > The only reason I do not raise further is that this change seems rather significant and thus, in my mind, requires another round of review.

---

### Official Review · Reviewer_9LtG · 2021-11-01

**Correctness:** 3
**Technical Novelty And Significance:** 3
**Empirical Novelty And Significance:** 3
**Recommendation:** 5
**Confidence:** 3

**Main Review:**

In general, I enjoy the paper while as every MDP paper has an unlimited preliminaries dense section.
I strongly believe that I will praise with large acceptance the first paper that will provide a more intuitive way of description of the RL/MDP notions. Of course this is not an issue of this specific problem but of the whole community.

So, let's go to my main questions

Issue #1. I feel that there is a huge lack of motivation in the problem. Can you give three nowadays good examples of Leader/Followers General-Sum MDP games in practice?

Issue #2. What are the novel parts of your algorithms? More precisely, what are the intuitive steps of the proposed algorithms and your novel introductions in a typical MDP scheme? I would like a extent answer in this issue because honestly I felt that the proposed algorithms were just plug-and-play folklore tools that we employ in practice.

**Summary Of The Paper:**

The paper studies multi-player general-sum Markov games with one of the players designated as the leader and the rest regarded as the followers. In particular, we focus on the class of games where the state transitions are only determined by the leader's action while the actions of all the players determine their immediate rewards.

**Summary Of The Review:**

Interesting paper with important theoretical contributions. There are issues about the motivation of the model.

---

> ### Author Response · Authors · 2021-11-19
> **To Reviewer 9LtG**
>
> Q1: I feel that there is a huge lack of motivation in the problem. Can you give three nowadays good examples of Leader/Followers General-Sum MDP games in practice?
>
> A1:  (1) Mechanism design (Conitzer & Sandholm, 2002; Roughgarden, 2004; Garg & Narahari, 2005; Kang & Wu, 2014): The leader imposes a price on each unit of bandwidth providing to each downloader. Then, the followers determine their optimal download bandwidths to maximize their individual utilities based on the assigned bandwidth price. (2) Security games (Tambe, 2011; Korzhyk et al., 2011): In this setting, a defender (leader) commits to a randomized deployment of security resources, and an attacker (follower) bestresponds by attacking a target that maximizes his utility. (3) Incentive design (Zheng et al., 1984; Ratliff et al., 2014; Chen et al., 2016; Ratliff & Fiez, 2020): There is a coordinator (planner/leader) and $n$ non–cooperative strategic agents (followers) each providing response. The followers play according to a Nash equilibrium strategy, and the goal of the leader is to design a mechanism to coordinate the followers by incentivizing them to take actions that ultimately lead to the minimization of objective functions. (4) We also give a simplified example in the introduction, which consists of a government and a group of companies. When the transition is determined by the current state and the action of government, this is a leader-controller general sum game.
>
> Q2: What are the novel parts of your algorithms? More precisely, what are the intuitive steps of the proposed algorithms and your novel introductions in a typical MDP scheme? I would like an extended answer to this issue because honestly, I felt that the proposed algorithms were just plug-and-play folklore tools that we employ in practice.
>
> A2: We want to emphasize that (1) the Stackelberg game is a bilevel optimization problem, and (2) the rewards depend on all players’ actions, and thus leader’s policy should take the followers’ best response into consideration when improving her policy own policy. Hence our setting is much harder than the single-agent MDP. To tackle these challenges, we propose a novel algorithm that adopts the optimistic estimation of leader’s value function for exploration and solves the bilevel optimization problem for policy optimization.
> Markov games also have some technical challenges even with the leader-controller assumption. For example, we need to use the $\epsilon$-SNE subroutine due to some technical issues. See the paragraph below Assumption 3.2 for details.

---

### Official Review · Reviewer_JEih · 2021-11-02

**Correctness:** 3
**Technical Novelty And Significance:** 3
**Empirical Novelty And Significance:** Not applicable
**Recommendation:** 5
**Confidence:** 3

**Main Review:**

The paper requires a considerable revision and re-organization. There are many details and remarks, but the setting, the challenges and the proposed solutions are unclear.

- First of all, the notion of linear function approximation is unclear. In Algorithm 1 (Line 10) and Algorithm 3 (Line 9), it is seen that the algorithm keeps and updates the value function for each state $x\in\mathcal{S}$ in an $|\mathcal{S}|$-dimensional array. We observe the same thing for the policies also, i.e., Line 8 in Algorithm 3 indicates that policies are kept for all $x\in\mathcal{S}$. This looks tabular, and for large state spaces, it is intractable. If there is an underlying parameterization, then storing and updating policies and value functions at each iteration is inefficient, the algorithms should indicate that only the corresponding value function (and policy) is computed. This should be clarified.

- The paper closely follows (Jin et al., '18; Jin et al., '20b) and the technical contributions seem a little incremental.

- The algorithm requires a matrix inversion at each iteration, and requires solving Line 2 of Algorithm 3, as the authors also indicate. I understand that the paper focuses on sample efficiency, but computational feasibility is an important concern especially in RL, especially in the function approximation setting where there is a large-scale problem. Could it be possible to solve the least-squares problem in Equation (3.2) by using SGD? How would that impact the sample complexity?

- What is the prevalence of the problem that is considered in this paper? There is a structural assumption on the transition dynamics, i.e., the transition kernel is linear with a given kernel. How broad is this class of problems?

- The literature review and notation are deferred to the supplementary material, and the paper is missing a conclusion. I think some space can be saved for these important discussions by deferring some of the technical details to the supplementary material instead.

- The paper considers structured (i.e., linear) Markov games, which is stated in Section 2.1. This important aspect can be emphasized for the sake of clarity earlier, perhaps in the abstract also.

**Summary Of The Paper:**

The paper investigates leader-controlled multi-player general-sum games in the episodic setting. Based on optimistic and pessimistic LSVI, the paper proposes algorithms that achieve sublinear regret in the online and offline setting, respectively.

**Summary Of The Review:**

The problem setting, the algorithms and the results require considerable clarification. The proposed methods can have large memory and computational complexities, which may be problematic for large state spaces. The technical contributions seem a little incremental.

---

> ### Author Response · Authors · 2021-11-19
> **To Reviewer JEih**
>
> Q1: First of all, the notion of linear function approximation is unclear. In Algorithm 1 (Line 10) and Algorithm 3 (Line 9), it is seen that the algorithm keeps and updates the value function for each state $s \in \mathcal{S}$ in an $|S|$-dimensional array. We observe the same thing for the policies also, i.e., Line 8 in Algorithm 3 indicates that policies are kept for all $x \in \mathcal{S}$. This looks tabular, and for large state spaces, it is intractable. If there is an underlying parameterization, then storing and updating policies and value functions at each iteration is inefficient, the algorithms should indicate that only the corresponding value function (and policy) is computed. This should be clarified.
>
> A1: This is wrong. (i) In Algorithm 1 (Line 10), we only need to store parameters $w_h^k$ and $\Lambda_h^k$.  (ii) In Algorithm 3 (Line 9), we only need to calculate the value and policy for the states encountered by the algorithm. So the computational cost is $\text{poly}(H)$. (iii) We also want to emphasize that our regret bound is independent of $|S|$ and $|A|$.
>
> Q2: The paper closely follows (Jin et al., '18; Jin et al., '20b) and the technical contributions seem a little incremental.
>
> A2: The algorithms in (Jin et al., '18; Jin et al., '20b) and most existing (provably efficient) single-agent RL algorithms are UCB-type algorithms. The core idea is adding bonus functions to promote exploration and thus achieve optimism. Our algorithms, variants of LSVI algorithms, also add bonus/penalty functions to guarantee optimism/pessimism. However, extending the result in single-agent MDP to multi-player general-sum games is highly non-trivial because the followers’ actions also affect the leader’s reward (this is not considered by (Jin et al., '18; Jin et al., '20b)). We first establish the $\tilde{O}(\sqrt{T})$ regret for general-sum Markov games.
>
> Q3: The algorithm requires a matrix inversion at each iteration, and requires solving Line 2 of Algorithm 3, as the authors also indicate. I understand that the paper focuses on sample efficiency, but computational feasibility is an important concern especially in RL, especially in the function approximation setting where there is a large-scale problem. Could it be possible to solve the least-squares problem in Equation (3.2) by using SGD? How would that impact the sample complexity?
>
> A3: This is a good question. Solving the least-squares problem in Equation (3.2) by using SGD may be a more efficient method in practice. However, using SGD will not admit the closed-form solution in Equations (3.3) and (3.4), and thus make the theoretical analysis intractable. To our best knowledge, existing work (e.g. [1], [2]) with regret guarantees all requires a matrix inversion at each iteration.
>
> We also remark that [3] did SDG for linear bandits, the same analysis can be used to study RL problems with parametric transition models, such as matrix bandit [4].
>
> [1] Chi Jin, Zhuoran Yang, Zhaoran Wang, and Michael I Jordan. Provably efficient reinforcement learning with linear function approximation. In Conference on Learning Theory, pp. 2137–2143. PMLR, 2020b.
>
> [2] Andrea Zanette, Alessandro Lazaric, Mykel Kochenderfer, and Emma Brunskill. Learning near-optimal policies with low inherent bellman error. In International Conference on Machine Learning, pp. 10978–10989. PMLR, 2020b.
>
> [3] Chi-Ning Chou, Juspreet Singh Sandhu, Mien Brabeeba Wang, and Tiancheng Yu. A General Framework for Analyzing Stochastic Dynamics in Learning Algorithms. arXiv preprint arXiv:2006.06171, 2020.
>
> [4] Lin Yang and Mengdi Wang. Reinforcement learning in feature space: Matrix bandit, kernels, and regret bound. In International Conference on Machine Learning, pp. 10746–10756. PMLR, 2020.

---

> ### Author Response · Authors · 2021-11-19
> **To Reviewer JEih**
>
> Q4: What is the prevalence of the problem that is considered in this paper? There is a structural assumption on the transition dynamics, i.e., the transition kernel is linear with a given kernel. How broad is this class of problems?
>
> A4: (1) Mechanism design (Conitzer & Sandholm, 2002; Roughgarden, 2004; Garg & Narahari, 2005; Kang & Wu, 2014): The leader imposes a price on each unit of bandwidth providing to each downloader. Then, the followers determine their optimal download bandwidths to maximize their individual utilities based on the assigned bandwidth price. (2) Security games (Tambe, 2011; Korzhyk et al., 2011): In this setting, a defender (leader) commits to a randomized deployment of security resources, and an attacker (follower) bestresponds by attacking a target that maximizes his utility. (3) Incentive design (Zheng et al., 1984; Ratliff et al., 2014; Chen et al., 2016; Ratliff & Fiez, 2020): There is a coordinator (planner/leader) and $n$ non–cooperative strategic agents (followers) each providing response. The followers play according to a Nash equilibrium strategy, and the goal of the leader is to design a mechanism to coordinate the followers by incentivizing them to take actions that ultimately lead to the minimization of objective functions. (4) We also give a simplified example in the introduction, which consists of a government and a group of companies. When the transition is determined by the current state and the action of government, this is a leader-controller general sum game.
>
> Linear transition kernel can be easily extended to RKHS [5]. RKHS is a broad class of functions. So linear transition is a reasonable assumption.
>
> [5] Zhuoran Yang, Chi Jin, Zhaoran Wang, Mengdi Wang, and Michael I Jordan. Bridging exploration and general function approximation in reinforcement learning: Provably efficient kernel and neural value iterations. arXiv preprint arXiv:2011.04622, 2020.
> Q5: The literature review and notation are deferred to the supplementary material, and the paper is missing a conclusion. I think some space can be saved for these important discussions by deferring some of the technical details to the supplementary material instead.
>
> A5: We have moved some related works on learning Stackelberg games to the main paper in the revision. Thanks for your advice.
>
> Q6: The paper considers structured (i.e., linear) Markov games, which is stated in Section 2.1. This important aspect can be emphasized for the sake of clarity earlier, perhaps in the abstract also.
> A6: We have emphasized this in the second line of the abstract. “In particular, we focus on the class of games where the state transitions are only determined by the leader’s action while the actions of all the players determine their immediate rewards.” We also emphasized this in the introduction. For example, the main question in the introduction is  “Can we develop reinforcement learning methods that provably find Stackelberg-Nash equilibria in leader-controlled general-sum games with sample efficiency?”

---

### Official Review · Reviewer_MweZ · 2021-11-03

**Correctness:** 3
**Technical Novelty And Significance:** 3
**Empirical Novelty And Significance:** 2
**Recommendation:** 5
**Confidence:** 2

**Main Review:**

The paper seems to consider the interesting problem of computing SNE for linear MDPS, which seems to be a novel problem. The technical level of the paper seems to be of high quality. However, the structure of the paper would benefit from some adjustments. In particular, the related work section is of paramount importance in this work, as it is not obvious how much of the presented work is already in the literature. I would suggest moving the related works section to the main paper. Also, the algorithms presented in the paper are hard to follow: it is hard to understand the meaning of instantiating a function.

The main idea of the work is not well transmitted. Since the case of H=1 is already solved in the literature, does your method just recursively use such oracle recursively on a game with optimistic estimates? What is the main difficulty of computing such equilibrium, w.r.t. the case H=1?

The paper would greatly benefit from an empirical evaluation of the algorithm. As a benchmark, I think you could employ a solver of normal form Stackelberg games on the normal form formulation of the sequential game. This could be done in small games. Since such analysis is expected a motivation on why it is missing would be welcomed. The related work section of the work misses a work that the reviewer is familiar with, namely: Balcan et al. "Commitment Without Regrets: Online Learning in Stackelberg Security Games" (2015). In that case, the horizon is H=1 and there are no externalities (the best response to the leader is a joint NE for the followers). However, the paper underlines that the problem poses many difficult features, such as non-convexity, and non-continuity of the payoffs, and, hence, a careful analysis of the strategy space is needed. In this work, such specific analysis seems missing. A discussion around this topic is hence welcomed.


**Summary Of The Paper:**

The paper studies the problem of multi-agent Markov games with one leader and multiple followers. Moreover, the games considered are controlled by the leader, meaning that the transition function only depends on the action of the leader. The goal is to find the Stackelberg-Nash equilibrium of the game by means of reinforcement learning-based algorithms, both for the offline and online setting. The algorithms proposed have guarantees in terms of regret which is sub-linear in the number of episodes for the online setting and converges to zero in the offline setting in the size of the dataset.

**Summary Of The Review:**

The paper would benefit from more discussion on the related works and analysis similar to the one presented in the work by Balcan (2015).

---

> ### Author Response · Authors · 2021-11-19
> **To Reviewer MweZ**
>
> Q1: the related work section is of paramount importance in this work, as it is not obvious how much of the presented work is already in the literature. I would suggest moving the related works section to the main paper.
>
> A1: We have moved some related works on learning Stackelberg games to the main paper in the revision. Thanks for your advice.
>
> Q2: the algorithms presented in the paper are hard to follow: it is hard to understand the meaning of instantiating a function.
>
> A2: What is the meaning of instantiating a function?
> Our algorithm is an optimistic variant of LSVI. In each round, we add a bonus function to the estimated Q-function for promoting exploration. Then, we calculate the policies by using LP or oracle to solve the normal-form game. Notably, we use an $\epsilon$-SNE subroutine because of some technical issues (see the paragraph below Assumption 3.2).
>
> Q3: The main idea of the work is not well transmitted. Since the case of H=1 is already solved in the literature, does your method just recursively use such oracle recursively on a game with optimistic estimates? What is the main difficulty of computing such equilibrium, w.r.t. the case H=1?
>
> A3: We have mentioned that, when there is only one follower, the normal form game can be calculated by linear programming (LP). However, when there are multiple followers, this cannot be calculated efficiently. Thanks for pointing out the related work on the Stackelberg security game (with one follower), and we have cited this paper in the revision.
> As for computation, the algorithm indeed recursively uses the one-step optimization oracle, which shows that our method can be efficiently implemented.
> Meanwhile, we want to emphasize that our focus is the Markov games setting, which involves state transition. We establish the $\sqrt{T}$ regret in this setting (first result). To achieve sample efficiency, we need to balance the tradeoff between exploration and exploitation in the context of Stackelberg games.

---

### Comment · Area_Chair_aD3A · 2021-11-20
**Please read responses from the authors**

Dear reviewers,

Please read the detailed responses from the authors. How do they change your evaluation? Do you still have major concerns? Thank you.

---

### Decision · Program_Chairs · 2022-01-20

**Decision:**

Reject

**Comment:**

This paper studies a Markov game with a single leader and multiple followers, where the state transitions are independent of the actions of the followers.  The paper studies online and offline RL methods for this subclass of the Markov game, and establishes a sublinear regret bound for the online RL method and sublinear suboptimality of the offline RL methods.

Although the paper appears to contain interesting ideas and contributions, the responses and revisions have not sufficiently addressed some of the major concerns of reviewers.  The reviewers and AC thus agree that the paper is not ready for publication.  For example, the issue of motivation and the myopic-follower setting has not been resolved yet.  Also, the discussion on the issues studied in Balcan et al. (2015) is not provided.  Please also note that (Jin et al., '20b) also proposes an optimistic variant of LSVI, and the exact algorithmic contributions still have some unclarity.